# Measuring and Modelling Structural Colours of *Euphaedra neophron* (Lepidoptera: Nymphalidae) Finely Tuned by Wing Scale Lower Lamina in Various Subspecies

**DOI:** 10.3390/insects14030303

**Published:** 2023-03-21

**Authors:** Zsolt Bálint, Gergely Katona, Szabolcs Sáfián, Steve Collins, Gábor Piszter, Krisztián Kertész, László Péter Biró

**Affiliations:** 1Hungarian Natural History Museum, Department of Zoology, Baross utca 13, 1088 Budapest, Hungary; 2Institute of Technical Physics and Materials Science, Centre for Energy Research, 29-33 Konkoly Thege Miklós St., 1121 Budapest, Hungary; 3African Butterfly Research Institute, P.O. Box 14308, Nairobi 00800, Kenya

**Keywords:** Adoliadini, Africa, biogeography, cline, reflectance spectrum, subspecies, thin film

## Abstract

**Simple Summary:**

Butterflies are best known because of their coloured wings. The colours are not for decorative but for crucial purposes within their reproductive strategy. One of these is sexual signalling, where the colours used for communication prior to mating are not chemical (based on pigments), but physical (based on nano-sized architectures). The genus *Euphaedra* belongs to the most diverse butterfly genera in forests of the Afrotropical Region, where many species are amazingly coloured. However, in the narrow forest belt along the Indian Ocean coast, *Euphaedra* is represented by only two species, one of them is the gold-banded forester (*E. neophron*) which expresses various physical colours. We investigated the coloration mechanism of this species, with our findings confirming that their structural colours are generated by the thickness of the laminae of the individual scales covering the wing surfaces. The colours are characteristic for all geographically distinct populations, currently recognised by scientists as subspecies. However, these do not show any clinal pattern or reflect climatic changes; therefore, we speculate that the various populations respond to new variables randomly tuning their colours.

**Abstract:**

The nymphalid butterfly *Euphaedra neophron* (Hopffer, 1855) is the only structurally coloured species representing the genus along the Indian Ocean coast in East Africa and Southern Africa, with a distribution from southern Somalia to the Kwa-Zulu-Natal region of South Africa. The range of *E. neophron* is subdivided to several, geographically distinct populations, currently recognised as subspecies by taxonomists on the basis of violet, blue, and green-coloured morphs. We investigated the optical mechanism of all these morphs by various materials science techniques. We found that the structural colour is generated by the lower lamina of the cover scales and the different colours are tuned according to their thickness, which was also proved by modelling. The colour tuning of the different subspecies does not reflect any clinal pattern, be it geographical or altitudinal.

## 1. Introduction

The subfamily Limenitidinae of Nymphalidae is a cosmopolitan group of true butterflies (Papilionoidea) inhabiting all faunal regions [1]. In the Old World and Australasian tropics, Adoliadini is the most characteristic tribe, often taking part in various mimicry rings composed around danaine and/or acraeine butterflies [2,3] and aposematic day-flying moths, such as the bright red, black and white-coloured geometrid group of *Scopula* (=*Aletis*). In the Afrotropics, especially the adoliadine genera *Bebearia* and *Euphaedra* display wide phenotypic diversity; hence, their taxonomy is not well understood [2,3,4]. Especially *Euphaedra* species are known for their “rainbow” colouration, as colours from almost black deep violet (*E. uganda* Aurivillius, 1895) and flashy green (*E. rubrocostata* (Aurivillius, 1898)) via highly reflective golden (*E. adonina* (Hewitson, 1865)) to fire red (*E. orientalis* Rothschild, 1898) can be seen and measured on their dorsal wing surfaces.

Regarding sexual dimorphism, which grossly influences colouration in butterflies [5,6,7], *Euphaedra* show also an unusually wide array of plasticity [2,4]. There are many species without remarkable difference between sexes (*E. orientalis*), but there are species with less or more pronounced sexual dimorphism (*E. crossei* Sharpe, 1902) to the extent that different morphs were recognized first as distinct species, as in the case of *E. medon* (Linnaeus, 1763). This phenomenon suggests that colours are differently used by the various species.

In this paper, we attempt to study a polytypic *Euphaedra* species distributed in the narrow forest belt along the Indian Ocean coast of Africa and in adjacent, sometimes disjunct montane and riverine forests: *E. neophron* (Hopffer, 1855). Regarding adult size, sexual dimorphism, and pattern, this species shows homogeneity throughout its distribution, but in terms of colouration, it displays geographical variability. Via applying various materials science techniques, our aims are (1) to document and describe the wing-scale morphology and the colour-generating mechanism of the species, (2) to characterize and understand the physics of the structural colours displayed by geographically distinct populations, and (3) to discuss briefly our results in the light of different topics such as diversification, former observations, and morph colour-tuning. With this paper, we provide an insight into the physical and biological background of the *Euphaedra* rainbow colours.

## 2. Materials and Methods

### 2.1. Species and Specimens

Classification of the investigated species *Euphaedra neophron* (Hopffer, 1855): Insecta: Lepidoptera: Nymphalidae: Limenitidinae: Adoliadini: *Euphaedra* Hübner, (1819) (type species: *Papilio cyprissa* Cramer, (1775); subsequent designation of Scudder in 1875). *E. neophron* was described by a detailed morphological account by Larsen [8], catalogued by Ackery et al. [9] and documented in full colour by D’Abrera [2]. Various morphs have been discriminated taxonomically, and named as subspecies [4] (Figure 1 and Figure 2; Table 1). Specimens used for experiments were curated in the African Butterfly Research Institute (Nairobi, Kenya) and in the Hungarian Natural History Museum (Budapest, Hungary). Their data and taxonomic identity are given in Table 2.

All specimens selected for examination were in good condition, with no visible wing damage. They represent the typical morph of the respective subspecies, none of them was an aberrant individual. For demonstrating the species-specific character of the spectra measured, we investigated four *E. n. violaceae* male specimens.

### 2.2. Optical Microscopy

Optical imaging of the wing scales was carried out using a Nikon Eclipse LV150N (Shinagawa, Tokyo, Japan) microscope in reflected light. The wing scales stand at an angle of approximately 15° respective to the wing membrane; so, for better visibility, we used focus stacking to compensate for the narrow depth of field of the high-resolution microscope objectives. Single-scale images were also taken in reflected light.

### 2.3. Scanning Electron Microscopy (SEM)

SEM images were taken using a Zeiss LEO (Jena, Germany) 1540 XB FESEM device. Wing pieces were cut and mounted on the sample holder with double-sided conductive tape; also, single scales were placed on conductive tape. To ensure that the original structure of the wing scales was preserved, no other treatment was applied.

### 2.4. Spectroscopy

We measured reflectance spectra in the forewing dorsal surface median area below the cubital vein with a modular Avantes (Apeldoorn, The Netherlands) fibre optic spectrophotometer (HS-1024*122TEC) and a light source (AvaLight DH-S-BAL). All spectra were recorded against a white reference standard WS-2 (Avantes). For the measurement of the reflected specular component, we used a bifurcated probe oriented normally (perpendicular) to the wing.

### 2.5. Modelling

The lower laminae of the wing scales were modelled as chitin thin films using a transfer matrix method [10,11]. The thickness of the lower lamina of the scales was measured on the SEM images of the cross-sections in multiple points. These values were averaged and used as input in the calculations. We simulated reflectance spectra for normal incident light and compared the results with the measured spectra. The calculated and measured results were transformed into the CIE 1931 chromaticity diagram, which represents the human colour vision and can be used for the comparison of reflectance spectra and for comparing the visual impression of colour perceived by a human observer.

## 3. Results

### 3.1. Scale Structures

The colour-generating scales of *Euphaedra neophron* in both sexes show the typical papilionoid ground plan [12,13]: both wing surfaces are densely covered by two layers of scales. In the dorsal surface, the scales of both layers are shovel-shaped, but the scales of the two layers differ slightly in shape and size. The ground scales are somewhat larger and shorter with undulated apices, and they are brown in reflected light. The cover scales are slightly longer but narrower, their apices are scalloped with three or four pointed termini and, in reflected light, they generate different hue of structural colours (Figure 3).

Regarding microstructures of the colour generating scales of all investigated subspecies, their dorsal surfaces are structured by longitudinal ridges and cross ribs, forming more or less rectangular ~2 × 1.5 μm windows. The longitudinal ridges are formed by non-*Morpho*-like narrow lamellae, the cross ribs are unstructured. The longitudinal ridges and the cross ribs are attached by trabeculae to the scale laminae. The surface of the scale lamina in both layers is flat, unstructured (Figure 4).

### 3.2. Spectral Properties

The four male specimens of *E. n. violaceae* share a spectrum with a main peak around 350 nm with a reflectance maximum between 18 and 28%. This reflectance peak is just below the range of human visual sense; therefore, the colour of *E. n. violaceae* seems to be somewhat faded violet. Above 550 nm, the reflectance increases according to the characteristic absorbance of melanin pigment present in the lumen of the scale [14]. The measured reflectance intensity of the male *E. n violaceae* varies but it is between the boundaries identified experimentally in case of other butterflies [6]. The shape of the female spectrum (#8) is similar with that of the males, one may remark the notable difference that the peak is shifted to 420 nm (Figure 5A).

The spectrum of *E. n. ellenbecki* is exceedingly similar to that of *E. n. violaceae* males; identical with the least reflective *E. n. violaceae* specimen. While using normal incidence measurements, the reflectivity differences may result by the different number of colour- generating cover scales due to mechanical incompleteness of the wing (Figure 5B).

The morph *E. n. littoralis* displays higher reflectivity compared to the previously mentioned taxa, and it is better observable also for the human eye because *E. n. littoralis* has the strongest blue colour amongst the examined specimens. The *E. n. littoralis* spectrum is very similar to that of *E. n. violaceae* female, with the difference that the main peak is shifted 25 nm to the red (Figure 5B).

The spectra of *E. n. neophron* and *E. n. zambesia* have their strong reflectivity more in the centre of the human eye spectral range: around 480 nm (*E. n. neophron*) and 550 nm (*E. n. zambesia*), with the result that the dorsal wing surface of *E. n. neophron* displays a metallic blue colour, whilst that of *E. n. zambesia* is green. With the highest reflectivity, the *E. n. neophron* colour is vivid and seems to be metallic for human perception. The main peaks are the widest amongst the investigated specimens. Their additional common peculiarity is that their spectra possess an additional peak in the ultraviolet, harmonic of the main peak. This trait is shared also by the female spectrum *E. n. violaceae*. This peak in the ultraviolet is also higher in *E. n. neophron* than in *E. n. zambesia*, which is also the case in the visible range (Figure 5B).

### 3.3. Modelling

One single layer of chitin with refractive index of n = 1.56 + 0.1i [15] in air (n = 1) ambient gives a reflectance as a function of the layer thickness. We calculated the reflectance in the range of 200–1200 nm, the main peak, and its harmonics in the UV shift to the red with the increasing layer thickness (Figure 6A). Seventeen spectra from Figure 6A were selected and plotted in CIE chromaticity diagram to form a continuous line (dashed line in Figure 6B). In addition, we plotted the measured spectra of butterfly wings as black dots.

Using SEM images of the cover scale cross-section (Figure 4C), it was possible to approximate the layer thickness of the lower lamina for the investigated specimens. Previously, it was shown that the structural colour of certain butterflies is determined by a single chitinous layer [16]. According to Figure 6A, a wide spectrum of structural colours can be generated by the tuning of the lamina thickness. Here, this is also the case, all investigated species show a reflectance spectrum correlated with the lamina thickness (Figure 7).

### 3.4. Morph Colour Tuning

There is a uniform appearance in *E. orientalis* across its entire range, while *E. neophron* displays structural colours which are characteristic for a given and well-defined geographical area. These colours generated by the different thickness of the scale lamina do not appear clinal since no geographical or ecological gradient correspond with the respective appearance of the distinct colour morphs from violet via blue to green and *vice versa*. If the colour diversity of *E. n. neophron* would be clinal, then the surface colour of the subspecies *E. n. meruensis* should correspond with the violet-coloured submontane *E. n. violacea*, showing a general darkening trend, expressing the sequence green-blue-violet from the south to the north. In this respect, the appearance of *E. n. zambesia* is also outstanding. Its green appearance contradicts the observations that populations occurring further inland from the coastal blue-coloured *E. n. littoralis* or *E. n. neophron* appear to be violet. This would suggest an east-west cline grade from blue to violet, but the presence of green *E. n. zambesia* contradicts the scenario. These findings are summarised in Figure 8, where the perceived hues of the structural colours were connected with the geographical locations of the populations, and no correlations were found.

## 4. Discussion

### 4.1. Former Observations

Larsen [8] monographing the butterflies of Kenya remarked that the colours of the subspecies *littoralis*, *meruensis*, and *violaceae* “… do not differ very strongly from each other, while the ground colour of ssp. *ellenbecki* is a beautiful golden brown with a hint of violet. Doubtless scale structure is responsible for the difference; probably the subspecies is ultra-violet rather than brown”. As we demonstrated, the spectral properties produced by the wings of *E. n. littoralis* and *E. n. violaceae* are different, which can be also detected by the perception of human eyes. The spectrum of *E. n. ellenbecki* is almost identical with that of male *E. n. violaceae*; but the origin of the “beautiful golden brown” colour impression of *E. n. ellenbecki* may be due to the lower density of colour-generating scales in the upper layer and the increasing reflectance above 600 nm (Figure 3).

### 4.2. Sexual Dimorphism

In the Introduction, it was mentioned that *E. neophron* displays no sexual dimorphism in colouration, which was also stressed by Larsen [8]. In spite of that, the spectrum of the female *E. n. violaceae* compared to the male spectra displays the notable difference that the peak in the female is shifted beyond 400 nm (Figure 5A). This confirms also the observation based on human eye, that is, the dorsal wing surface structural colour in the *E. n. neophron* group of *Euphaedra* amongst sexes is different, but in other groups, it is not (cf. [2]: 394–396). Hence, we can presume that the dorsal wing surface colouration in the *E. n. neophron* group is also a species-specific signal and may contribute for a better recognition for conspecific individuals (cf. [17,18]). As male and female behaviours are markedly different, the specially tuned structural colours result in an obvious signal for the opposite sexes.

Our results and observations on *Euphaedra* sexual dimorphism are based on the experimental investigation of four males and a single female specimen of *E. n. violaceae*. Therefore, they have to be taken with caution. It is important to test our findings in the cases of other subspecific morphs. However, it is not an easy issue, as female *E. neophron* individuals are rather rare in nature as well as in scientific collections.

### 4.3. Diversification and Colour Tuning Mechanism

It is well recorded now that various lineages of butterflies manipulate colour by changing architectures in the nano-realm of their scales covering the wing membrane, which can be modelled (cf. [19,20]). One of the easiest ways is to change the thickness of the scale lamina to achieve different structural colours. The application of this mechanism serves not only the diversification of *E. neophron* morphs, but also as a background for the colour diversity of the genus displaying by the various species.

To a more intuitive interpretation, we represented the experimental data and calculated the dataset in the CIE chromaticity diagram, based on human visual pigments. This shows the hue connected to a spectral dataset, and the hue of the wing surface of a corresponding species. The connecting line determines a path in the CIE diagram, crossing through all the possible hues of monolayered wing scales. The measured reflectance of the investigated specimens fell near this path, showing that the measured lamina thickness and the measured structural colours are well correlated. In detail, the measured reflectance of *E. n ellenbecki* and *E. n. violaceae* males were in a good agreement with the simulated spectra (Figure 7A) of the 180 nm and 185 nm thin films which were measured in SEM images, respectively. Furthermore, structural colours of *E. n. neophron* and *E. n. zambesia* males were also similar to the simulated spectra of SEM-based 275 nm and 240 nm chitinous laminae, respectively (Figure 7B).

### 4.4. Biogeographical Background

In spite of the high diversity of *Euphaedra* in sub-Saharan Africa’s tropical forests, in the Indian Ocean coastal forest area, there are only two species representing the genus: (1) *E. neophron* and (2) *E. orientalis*. The latter species is monotypic with pigment-coloured red wing surfaces; most probably, it belongs to a mimicry ring, which among multiple other butterfly species in four different families mimic the day-flying geometrid moth group in the genus *Scopula* (=*Aletis*): e.g., *Papilio dardanus* f. *trophonius* (Papilionidae), *Telipna sanguinea*, *Pseudaletis agrippina* (Lycaenidae), *Abantis eltringhami* (Hesperiidae). In contrary, *E. neophron* is polytypic with structurally coloured wing surfaces (Figure 1). This suggests different adaptation strategies for habitat occupation, possibly driven by different environmental variables: *E. orientalis* is able to use the potential tuned to various mimicry ring morphs, whilst *E. neophron* is able to handle fine-tuning of colours.

Indeed, there is growing evidence that living organisms, such as butterflies, are widely using the strategy to manipulate their body-covering exoskeletal parts in changing the nanoarchitectures for a better life management, which is obviously reflected by their colour display [21,22,23,24]. Certainly, this is the case also for the *Euphaedra neophron* subspecies formation, which indicates larger biogeographical events being the drivers for such diversification. These possibly correspond with the expansions and subsequent shrinkages of the tropical forest areas in East Africa due to global climatic shifts. Plant fossil data indicate that the tropical forest cover stretched across equatorial Africa during the late Oligocene, early Miocene between 30–25 Million years ago, but the forest area shrunk significantly after mid-Miocene, cc. 15–17 Million years ago [25]. Almost up to present time, forest habitats remained restricted to coastal mosaics or to the isolated mountainous areas in East Africa, such as the Eastern Arc Mountains in Tanzania, where forest was continuously present even during the last glacial maximum [26]. These isolated forest “refugia” could have been partially or fully (re)connected by habitat fluctuations occurring during the preceding wetter interglacial periods, as derived from fossils, marine pollen sediment, and sweet water pollen sediment or in the Holocene [27]. However, the recognised discreet taxonomic units, currently accepted as subspecies of *Euphaedra neophron* do not follow any recognised gradient or clinal pattern but responded to their respective environment completely individually (c.f. Figure 8).

### 4.5. Nomenclature

*E. neophron neophron* (Hopffer, 1855) inhabits South African coastal forest from South Africa (KwaZulu-Natal Region) via Mozambique to southern Tanzania, but it is also present further inland via isolated populations in forests on Mount Namuli, Mount Inago, Mount Mabu, and Mount Mecula (Mozambique) [28,29]. A couple of populations are still known from further inlands in Malawi (Mount Mulanje, Zomba Mountains), which appear differently with much greener metallic reflection as we demonstrated. For this population, we applied conditionally the available subspecific name “zambesia”. This name (*Romalaeosoma zambesia*, collection Felder syntype male(s) “Africa meridionalis, Zambesia”: [30]) hitherto was considered as synonym of *E. n. neophron*, but the type was never examined. If the type can be located (many Felder types are in the Natural History Museum, London, cf. [31,32]), then a decision can be achieved whether our application of the name was correct or our “zambesia” needs a new name.

### 4.6. Systematics

Classifying *Euphaedra*, Hecq proposed the subgeneric name *Neophronia* for *Euphaedra neophron* indicating the distinctness of the species [33]. Further investigations are necessary to map and analyse the connection between the various characters and molecular markers and to judge whether this subgeneric name is indeed necessary.

Another interesting question is the age of the subspecies recognized. The answer may put light on also whether the species *E. neophron* is comprised by some older lineages which beside *E. n. neophron* may be considered as species. For these topics, molecular markers would be necessary to explore for the differences of the various populations and to determine their genetic distances.

## 5. Conclusions

Using methods of materials science, we demonstrated that the various morphs of *Euphaedra neophron*, at present considered as different subspecies, produce their dorsal wing surface structural colour by a manner widely adopted by papilionoid butterflies: the lower lamina thickness of the scales covering the wing membrane is responsible for the colour generation by thin film interference. We did not found correlation between the colours and gradually changing climatic conditions, or geographic locations. Therefore, we concluded that the structural colour diversity of *E. neophron* populations reflects the responses of the investigated populations to environmental variables and the phenomenon cannot be considered as a clinal pattern.

## Figures and Tables

**Figure 1 insects-14-00303-f001:**
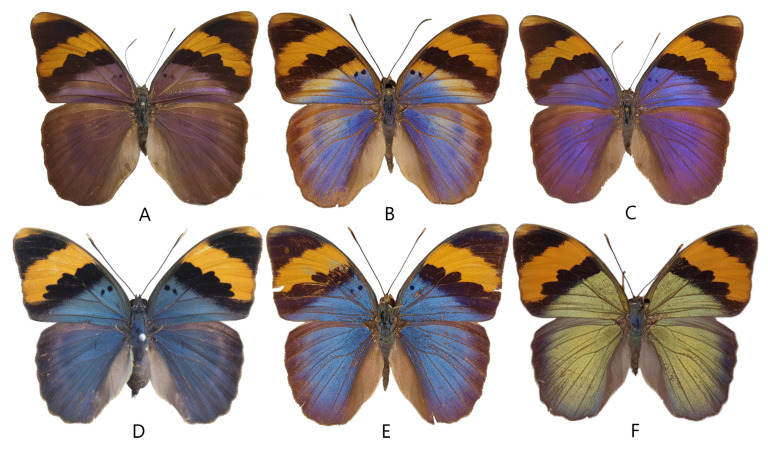
The main morphs of *Euphaedra neophron* subspecies from violet to green. The dominant colour is indicated as violet (V), blue (B), and green (G): **A** = *E. n. violaceae* (V); **B** = *E. n. ellenbecki* (V); **C** = *E. n. littoralis* (B); **D** = *E. n. rydoni* (B), **E** = *E. n. neophron* (B); **F** = *E. n. zambesia* (G).

**Figure 2 insects-14-00303-f002:**
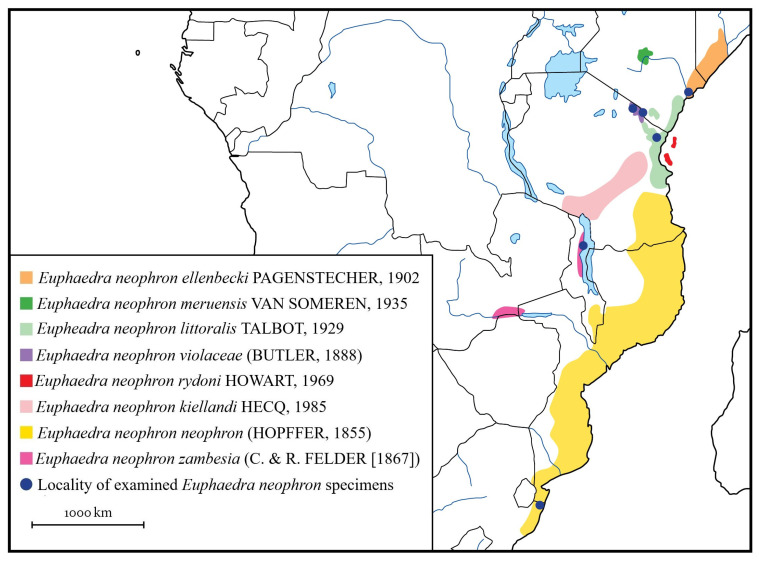
The distribution of hitherto recognised *Euphaedra neophron* subspecies (squares in various colours) and specimens examined (dark blue spots). State borders are indicated by black, rivers by light blue lines.

**Figure 3 insects-14-00303-f003:**
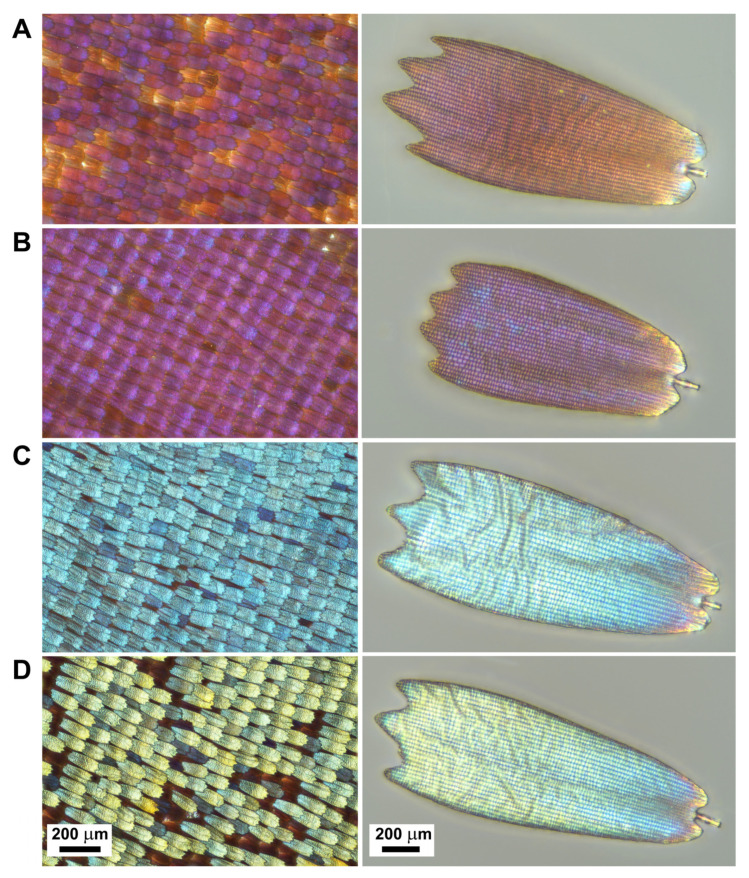
Optical micrographs of dorsal wing surface and an individual cover scale of various *Euphaedra neophron* morphs. (**A**) *E. n. ellenbecki,* (**B**) *E. n. violaceae,* (**C**) *E. n. neophron,* (**D**) *E. n. zambesia*.

**Figure 4 insects-14-00303-f004:**
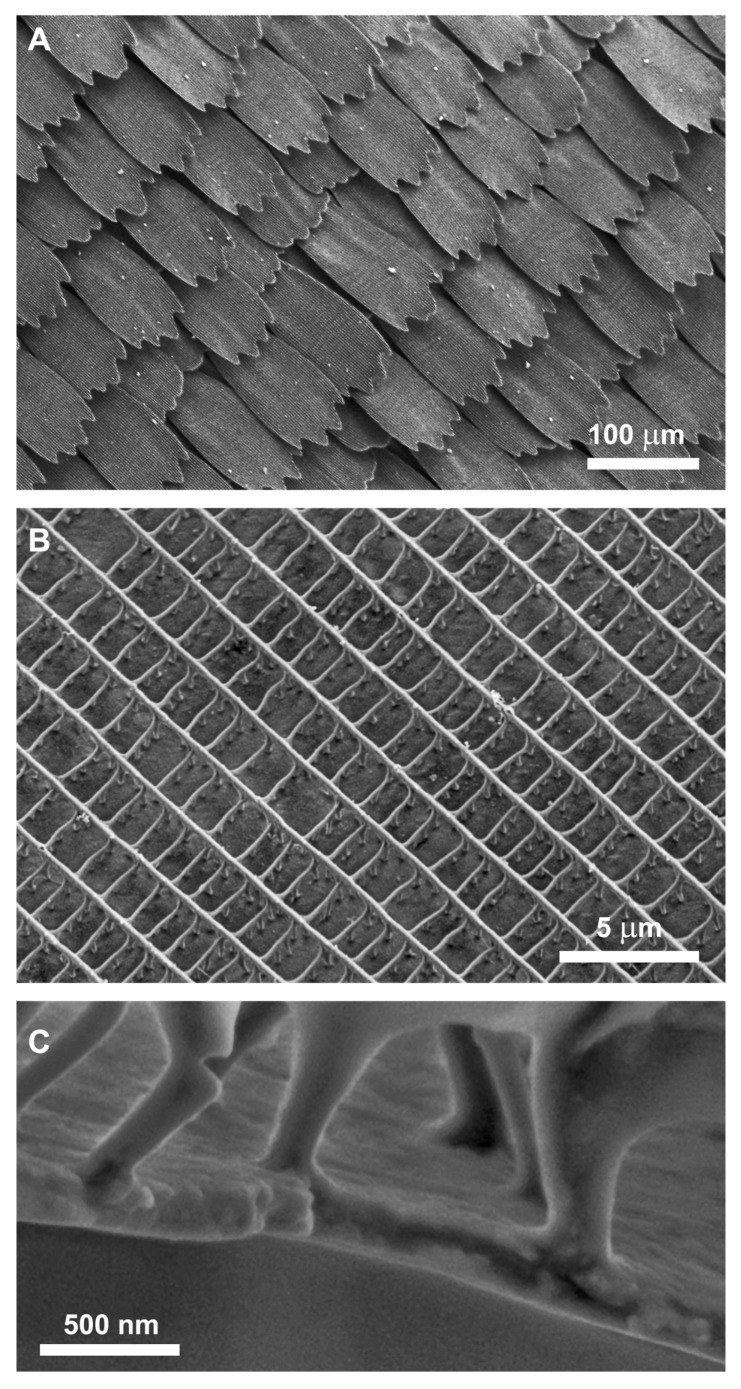
Scanning electron micrographs of *Euphaedra neophron violaceae* under different magnifications. (**A**) Dorsal hind wing scales generating violet colour, (**B**) longitudinal ridges and connecting cross ribs raising from the unstructured lamina of an individual scale of the region shown in (**A**). (**C**) Scanning electron micrograph of dorsal wing surface colour-generating scale cross-section, lamina, and trabeculae.

**Figure 5 insects-14-00303-f005:**
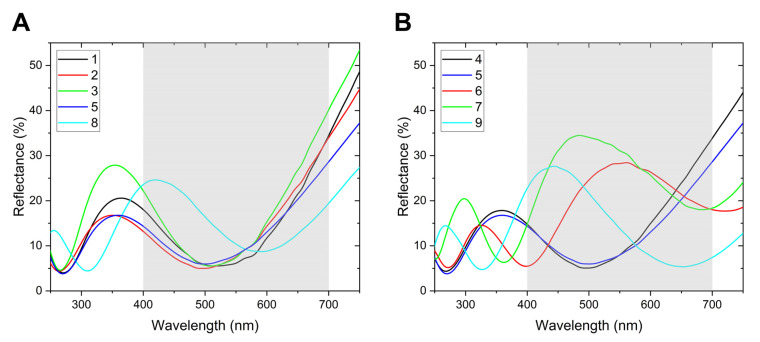
Reflectance spectra of the investigated *Euphaedra neophron* specimens (for working numbers, see Table 2). (**A**) 1, 2, 3, 5: *E. n. violaceae* males, 8: female. (**B**) 4: *E. n. ellenbecki*, 5: *E. n. violaceae*, 6: *E. n. zambesia*, 7: *E. n. neophron*, 9: *E. n. littoralis.* The grey area on the graphs shows the spectral sensitivity range of the human eye. Specimen no. 5 appears also in graph (**B**) as reference.

**Figure 6 insects-14-00303-f006:**
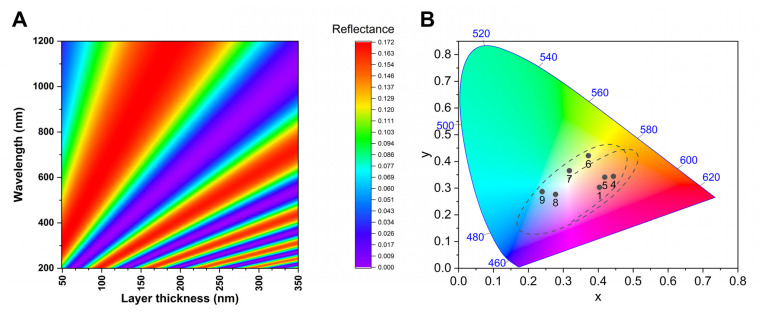
(**A**) Calculated reflectance spectra of single chitin layers of different thicknesses in *Euphaedra neophron* scales. (**B**) Chromaticity diagram showing the simulated data of (**A**) as a dashed line, and the reflectance of the measured butterfly wing surfaces: 1, 5, 8: *E. n. violaceae*, 4: *E. n. ellenbecki*, 6: *E. n. zambesia*, 7: *E. n. neophron*, 9: *E. n. littoralis* as black dots. The blue numbers show the position and wavelength of the monochromatic light.

**Figure 7 insects-14-00303-f007:**
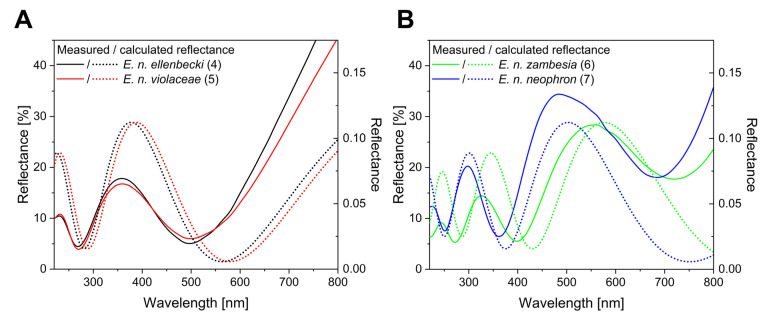
Measured and simulated reflectance spectra of *Euphaedra neophron* (with sample numbers): (**A**) *E. n. ellenbecki* (4) and *E. n. violaceae* (5); (**B***) E. n. zambesia* (6) and *E. n. neophron* based on reflectance measurements and SEM images, respectively.

**Figure 8 insects-14-00303-f008:**
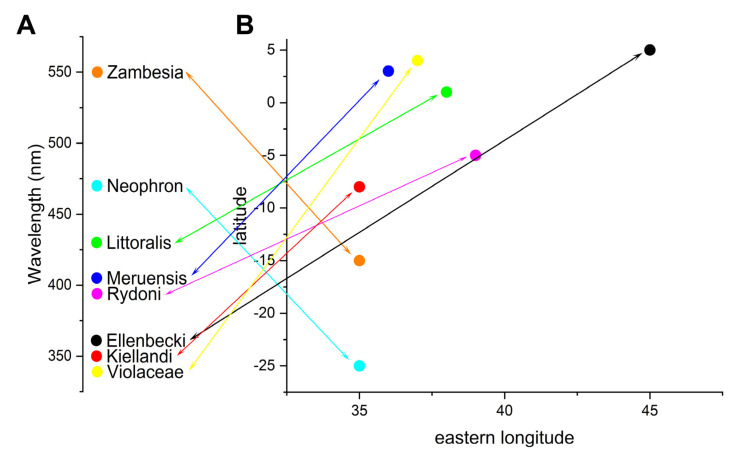
(**A**) Hypothetical grade of *Euphaedra neophron* subspecies colour tuning arranged on the basis of the peak wavelength. (**B**) The matching geographical positions of the populations. Data used for the compilation: *E. n. ellenbecki* = violet (360 nm), Somalia (5 N; 45 E); *E. n. kiellandi* = violet (350 nm), Tanzania (8 S; 35 E); *E. n. littoralis* = blue (430 nm), Kenya (1 N; 38 E); *E. n. meruensis* = blue (410 nm), Kenya (3 N; 36 E); *E. n. neophron* = blue (470 nm), Mozambique (25 S; 35 E); *E. n. rydoni* = blue (390 nm), Pemba Island (5 S; 39 E); *E. n. violaceae* = violet (340 nm), Kilimanjaro (4 N; 37 E); *E. n. zambesia* = green (550 nm), Malawi (15 S; 35 E). Coloured arrows connect the place of morphs on the peak wavelength representation and their geographical locality.

**Table 1 insects-14-00303-t001:** List of *Euphaedra neophron* subspecies arranged in alphabetical order. General geographical distribution is given. The type locality is indicated between quotation marks (with the actual political state affiliations).

Scientific Name	General Distribution	Type Locality
*Euphaedra neophron ellenbecki* (Pagenstrecher, 1902)	coast of northern Kenya and southern Somalia	“Umfudu” and “Evar” (Somalia)
*Euphaedra neophron kielland* (Hecq, 1985)	southern Tanzania	“Tanzanie, Ifakara, Musagabi” (= Ifakara, Masagati; Tanzania)
*Euphaedra neophron littoralis* (Talbot, 1929)	northern coast of Tanzania across the remaining patches of coastal of Kenya (to about Malindi)	“Rabai” (= Rabai Mpya (New Rabai); Kenya)
*Euphaedra neophron meruensis* (van Someren, 1935)	eastern slopes of Mount Kenya (Meru)	“Meru Forest” (Kenya)
*Euphaedra neophron neophron* (Hopffer, 1855)	coast regions from eastern South Africa via Mozambique to southern Kenya	“Mossambique” (Mosambique)
*Euphaedra neophron rydoni* (Howarth, 1969)	Pemba Island	“Pemba Island” (Tanzania)
*Euphaedra neophron violaceae* (Butler, 1888)	Usambaras and the South Pare Mountains in the Eastern Arc Mountains, also on Mount Meru and Mount Kilimanjaro in north Tanzania, and in Taveta, in the nearby Kenya	“Kilimanjaro” (Kenya and Tanzania)
*Euphaedra neophron zambesia* (Felder and Felder, 1865)	eastern Zimbabwe, Malawi, western Mozambique	“Zambia” (Zambia)

**Table 2 insects-14-00303-t002:** List of specimens examined. Abbreviations: ABRI = African Butterfly Institute, HNHM = Hungarian Natural History Museum.

Scientific Trinomen	Sex	Working Number	Original Data of the Specimen (Depositor)	Resolved Data (with Coordinates)
*Euphaedra neophron violaceae*	male	(1)	Kibosho, Kitty, coll. Velez; Tanzania, Mkoringa, leg. Kittenberger (HNHM)	(3°14′40.92″ S, 37°18′8.14″ E; altitude: 1281 m
*Euphaedra neophron violaceae*	male	(2)	Moshi, Kilimanjaro, 1936.IV.; coll. Velez (HNHM)	Tanzania, Moshi (3°20′40.89″ S, 37°20′50.59″ E; altitude: 800 m
*Euphaedra neophron violaceae*	male	(3)	Tanganyika, 1934. IV.; coll. Velez (HNHM)	Tanzania, Tanganyika (inaccurate locality)
*Euphaedra neophron violaceae*	female	(8)	Kenia, Kilimanjaro, 1913, VI.; coll. Velez (HNHM)	Kenya, Kilimanjaro (inaccurate locality)
*Euphaedra neophron violaceae*	male	(5)	Kenya, Kitobo Forest Taveta, 1977. III., S.C. Collins (ABRI)	Kenya, Kitavo (3°23′57.03″ S, 37°40′37.50″ E, altitude: 700 m)
*Euphaedra neophron ellenbecki*	male	(4)	Kenya, Witu, 400 ft, K 1983., S. C. Collins (ABRI)	(2°23′4.35″ S, 40°28′30.18″ E; altitude: 22 m)
*Euphaedra neophron zambesia*	male	(6)	Malawi, Nkhata Bay, 50 ft, 1983. III, S. C. Collins (ABRI)	(11°13′12.8″ S 34°16′44.5″ E; altitude: 500 m)
*Euphaedra neophron neophron*	male	(7)	South-Africa, Natal, Manguze, 1991.VII.12., A. Hanekom (ABRI)	South-Africa, Manguzi Forest (16°59′51.56″ S, 32°43′20.54″ E; altitude: 72 m)
*Euphaedra neophron littoralis*	male	(9)	Tanzania, East Usambara, forest below Camp Mawingu, 15.V.2022. Leg.: Sáfián, Sz. (HNHM)	(25°8′16.41″ S, 38°34′48.80″ E; altitude: 770 m)

## Data Availability

All relevant data analysed are included in this published article.

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
