# Peer review of "Measuring and Modelling Structural Colours of Euphaedra neophron (Lepidoptera: Nymphalidae) Finely Tuned by Wing Scale Lower Lamina in Various Subspecies"

_insects, 2023, doi:10.3390/insects14030303_

Round 1

Reviewer 1 Report

This study compares the color variation of various subspecies of E. neophron in Africa.  The authors approached this from various angles and using complementary techniques, including modelling.  The results are interesting and add important information to the field.  The text flows well and the figures are excellent.  My concerns, suggestions and queries are:

(1) From experience, scale morphology varies across the wing surface.  Thus, the authors should explain in the Methods which region of the wings was sampled for the morphological and spectral analyses (hopefully the same region for all specimens).

(2) Only one female specimen of E. n. violaceae was sampled, so the discussion of sexual dimorphism for the species should be viewed with caution.  Perhaps a simple mention that the spectral pattern in Fig. 5 would benefit from confirmation (examination of additional female specimens, and also female specimens of other subspecies).  Still considering Fig. 5, the reflectance curves for specimens 1, 2 and 5 are quite similar, while 3 has a higher reflectance peak at 350 nm.  Any idea why that was the case? 

(3) Any study that examines a single specimen of each taxon runs the risk of sampling “an unusual one” (as specimen 3, above).  It might be worth mentioning how the individual specimens were selected for the analyses. Furthermore, sampled specimens varied in collection date.  Authors could note that specimens number 2 and 5 (for example) were collected in 1936 and 1977, yet their spectral patterns were nearly identical (Fig. 5). 

Finally, there are some missing italics (lines 77 and 78 in the text, also some in the literature cited), and one typo (line 45 acreaine should be acraeine).

Reviewer 2 Report

Overall, I think this a well written manuscript that clearly describes the physical characteristics of scale structure that determine colour patterns in E. neophron. There are some odd English grammar use within the manuscript, which is likely related to authors being non-native English speakers. Not necessarily incorrect, but awkward for native English speakers. 

All of my comments below are minor.

L17 "...strategy for disseminating life." is an odd phrase selection. I assume the authors just mean reproductive strategy, which makes more sense. A reader could interpret the former as dispersal too.

L22 Typically insect common names are not capitalized, only if they include a proper noun.

L30-31 I would follow the same geographic direction in both lines. L31 would then be "...southern Somalia to Kwa-Zulu-Natal region of South Africa." Also, since they are not formal regions, but geographic descriptions, it should be east Africa to southern Africa.

L33 Since they are recognized as distinct subspecies, then maybe morphs would be a more appropriate term instead of phenotype. In reality, both are appropriate and this just may be minor semantics. 

L77 Make sure you italicize scientific names throughout.

Figure 2 The bodies of water and rivers are confusing as they are also coloured polygons but with no definition in the legend. They don't add to the map so they should just be omitted.

Table 1 isn't really formatted as a table.

L173 Authors should include binomial initials each time referencing the subspecies throughout - e.g., E. n. violaceae - makes it easier for the reader to follow, especially since one of the subspecies names is the same as the specific epithet.

211 Shouldn't start a sentence with a numeral. Should be spelled out - "Seventeen spetra from..."

Figure 7 Make note of the numbers after the subspecies names as the working number from Table 2.

Figure 8 This is introducing a new representation of the geographic and reflectance data. It should not occur in the discussion, this needs to be moved to the results.

Reviewer 3 Report

While I enjoyed reading the manuscript and looking at beautiful figures, I felt that it is of a limited value in its present form. I think that the subspecies category is very useful, but many people in the scientific community tend to reject it because it is used arbitrarily, especially by lepidopterists.

In the present study, the authors have an opportunity to contribute to our understanding of geographic speciation in butterflies by analyzing structural colors of different disjunct populations. But it seems that the manuscript can be improved as follows:

 Address the taxonomy of this group first: What makes the authors think that these populations represent the same species? The fact that they are treated as such by Larsen, or D’Abrera, or curated as such in the collection is not evidence. Do they interbreed in the contact zones (if any)? Are there intermediates? What is the DNA barcode differences between these populations? Are there any other molecular markers that have been explored? If the answers to these questions “we don’t know” then, please, sequence a few DNA barcodes from each of these isolated populations and provide genetic distances in the form of a matrix.

Also, please illustrate a male and a female for each taxon examined. In general, it would be helpful to have some statistical aspect to this paper. For each subspecies, there should be at least 3, but better 5 male individuals for which the reflectance spectrum is measured.

I encourage the authors to resubmit their paper after these points are addressed. It just needs a bit more work.
